# Reinforcement of Timber Dowel-Type Connections Using Self-Tapping Screws and the Influence of Thread Configurations

**Cong Zhang** [1,2], **Hao-Yu Huang** [3], **Xiong-Yan Li** [1,2], **Su-Duo Xue** [1,2], **Wen-Shao Chang** [4,*] **and Guo-Jun Sun** [1,2,*]

1    Faculty of Architecture, Civil and Transportation Engineering, Beijing University of Technology, Beijing 100124, China
2    The Key Laboratory of Urban Security and Disaster Engineering, Ministry of Education, Beijing University of Technology, Beijing 100124, China
3    School of Engineering, Newcastle University, Newcastle upon Tyne NE1 7RU, UK
4    School of Architecture, University of Sheffield, Sheffield S10 2TN, UK
*    Correspondence: w.chang@sheffield.ac.uk (W.-S.C.); sunguojun@bjut.edu.cn (G.-J.S.)

**Abstract:** The application of self-tapping screws as reinforcement on glulam connections has been proven effective. However, the implication of different thread configurations on the effectiveness of reinforcement remains unknown. This paper conducted experiments using screws with various thread configurations in embedment-strength tests and tensile connection tests. Results show that self-tapping screws with one third of thread achieved similar improvement in the embedment strength and mechanical properties of connections as fully threaded screws. This implies that properly reducing the thread length on self-tapping screws ensures easier screw installation than using fully threaded screws. The influence of screw-to-dowel distance was also investigated and two distances (0.5 d and 1 d) were adopted, with 'd' being the diameter of the dowel. The difference in embedment strength due to different screw-to-dowel distances was insignificant. The group with screws placed in contact (0.5 d) with the dowel achieved 5% higher embedment strength than the group with screws placed at a 1 d distance. The connection tests showed good agreement with the embedment-strength tests. This confirms that self-tapping screws with reduced thread can enhance the load-carrying capacity and ductility of connections to a level similar to connections reinforced by fully threaded screws.

**Keywords:** timber connections; dowel-type connections; reinforcement; self-tapping screws; load-carrying capacity; embedment strength

## 1. Introduction

Dowel-type connections have been widely used in large-scale glulam timber structures and timber–concrete composite structures [1]. Fluctuations in the relative humidity of the surrounding environment can lead to the formation of a moisture gradient and internal stresses within the timber elements. As moisture exchange within the wood occurs, dimensional change in the timber element is inevitable. However, the dimensional change is often restrained around the connection area, thus causing additional stresses. Due to the relatively low perpendicular to grain strength of the timber element, excessive stresses can lead to crack initiation and propagation around the connection area. The mechanical performance of a connection is usually critical, especially for timber structures in seismically active areas. The negative impact of cracks will place the mechanical performance of timber connections in question and affect the structural integrity of timber spatial structures. Therefore, various methods of reinforcement have been applied to improve the mechanical performance of timber connections.

Several studies [2–8] have used steel plates, nail plates and FRPs (fibre-reinforced polymers), respectively, as reinforcement to repair damaged timber members. However,

both reinforcement methods require a large amount of work and involve complex installation procedures. In addition, when such reinforcement is to be placed on timber members, accessibility to a large surface area of the structural components is usually required, which can limit their application when repairing certain historical buildings.

A trend in the latest studies shows that self-tapping screws are suitable and efficient connectors in modern glulam and CLT structures [9–14]. However, with the development of manufacturing technologies, the application of self-tapping screws has reached far beyond their traditional role as connectors. In recent decades, self-tapping screws have shown the potential to reinforce the timber perpendicular to the grain and prevent the splitting of wood around dowels. Blaß and Schmid [15] placed screws between fasteners in connections to enhance performance and the results have shown significant improvement in ductility. Bejtka and Blaß [16] and Blaß and Schädle [17] further reported that the load-carrying capacity and ductility of the connections were greatly enhanced when the screw was placed in contact with the dowel. Experimental studies by Lokaj and Klajmonová [18] on reinforcing round wood connections identified that screw reinforcement has the advantages of simplicity and low cost compared to other forms of metal reinforcement, such as steel plates. More recent studies on screw reinforcement further demonstrate its capability to enhance the mechanical performance of connections made from bio-based materials [19–25].

Within the scope of this research work, the authors improved the moment-resisting capacity of glulam connections using partially threaded screws on portal frames [26] and glulam connections with artificial cracks [27]. Screw reinforcement can control wood splitting and this ability is governed by the pull-through and withdrawal capacity of the screw and the similar 'rope effect' is discussed [28]. The capability of self-tapping screws to reinforce artificially damaged connections was also investigated [29]. Fully threaded screws require higher torque during installation. Thus, higher frictional force is produced during the installation process and increases the risk of damaging the screws, as discovered in [30]. Therefore, using self-tapping screws with reduced thread length is more beneficial for the purpose of both reinforcement and installation.

However, changing the thread length and thread location may influence the effectiveness of screw reinforcement, although limited knowledge is available. In addition, there are no design codes which specify the screw-to-dowel distance when reinforcing timber connections.

Therefore, this study, based on the previous work that was published in a shorter conference version [31], aims to understand the influence of thread configuration and screw-to-dowel distances on the effectiveness of screw reinforcement based on a series of embedment-strength tests, as embedment strength is an important factor in calculating the load-carrying capacity of connections in Eurocode 5 (EC5 hereafter) [32]. The following tensile connection tests were designed to validate the findings from embedment-strength tests.

## 2. Materials and Methods

### 2.1. Material Preparation

C24 European Whitewood (*Picea abies*) beams were used to prepare the specimens in this study. For the embedment and tensile connection tests, the specimens had an average density of 456 kg/m$^3$ (average moisture content of 9.2%) and 467 kg/m$^3$ (average moisture content of 11.6%), respectively. A Brennenstuhl moisture meter (Hugo Brennenstuhl GmbH, Germany) was used to obtain the moisture content of the specimen. A total of six measurements were taken for each specimen, to calculate the average moisture content. Details of the self-tapping screws are shown in Figure 1. To acquire screws with various thread configurations, part of the thread was removed by a grinder.

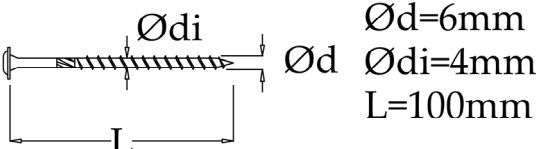

**Figure 1.** Flange-head partially threaded self-tapping screw used in this study.

## 2.2. Embedment-Strength Test Set-Up

The test set-up and specimen preparation, as shown in Figure 2, followed the European standard, BS EN 383:2007 [33]. A 2.5 mm diameter pre-drilled hole was drilled at the reinforcement location before the screws were driven in.

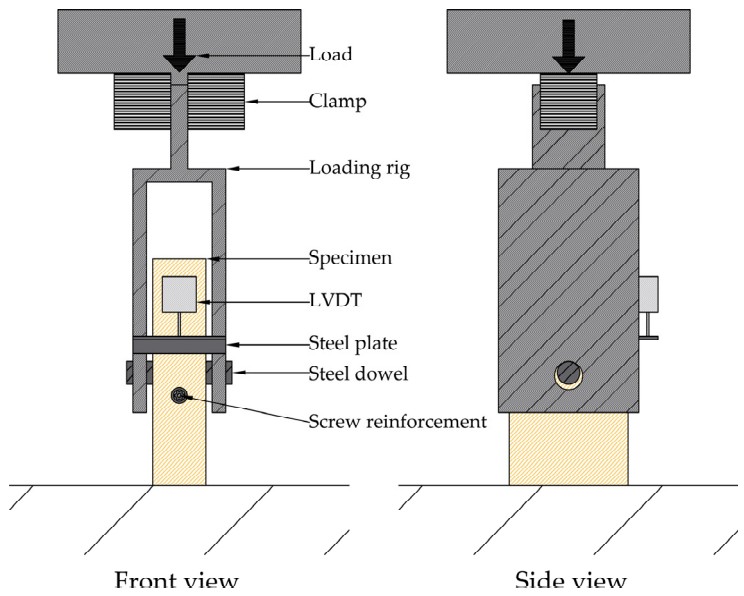

**Figure 2.** Embedment-strength test set-up.

A DARTEC 100kN loading machine (DARTEC LTD, West Midlands, UK) was used for the test. The loading was applied to the specimen parallel to the grain through a 20 mm diameter steel dowel and the displacement rate was set to 2 mm/min. The loading was stopped after 20% load drop from the peak load had been observed. Table 1 summarises the key information for each group in this study and Figure 3 demonstrates the thread configurations.

**Table 1.** Summary of each group in the embedment-strength test [31].

| Group | Description | Screw-to-Dowel Distance | No. of Specimens | Mean Density (kg/m$^3$) (CoV) | Mean M.C.% (CoV) |
|---|---|---|---|---|---|
| U | Unreinforced | N/A | 15 | 452 (9%) | 9.6 (20%) |
| N | 0% thread | 1d | 15 | 461 (8%) | 8.6 (21%) |
| S | 100% thread | 1d | 15 | 453 (6%) | 8.6 (18%) |
| BS | 33% thread on point end | 1d | 15 | 459 (5%) | 9.2 (20%) |
| DS | 33% thread on head end | 1d | 15 | 456 (9%) | 9.8 (14%) |
| ES | 100% thread | 0.5d | 15 | 452 (8%) | 9.5 (15%) |
| TTS | 33% thread on both ends | 1d | 15 | 459 (9%) | 9.3 (18%) |

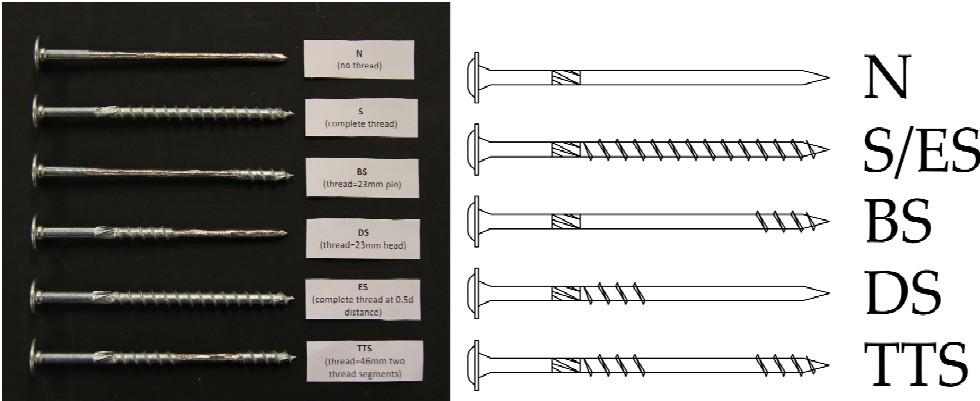

**Figure 3.** Screws with different thread lengths.

*2.3. Tensile-Connections Test Set-Up*

The design of the timber–steel–timber connections followed the guidance given by EC5. Details of the test set-up of the tensile connections are shown in Figure 4. A DARTEC 2000 kN loading machine (DARTEC LTD, West Midlands, UK) was used for the test.

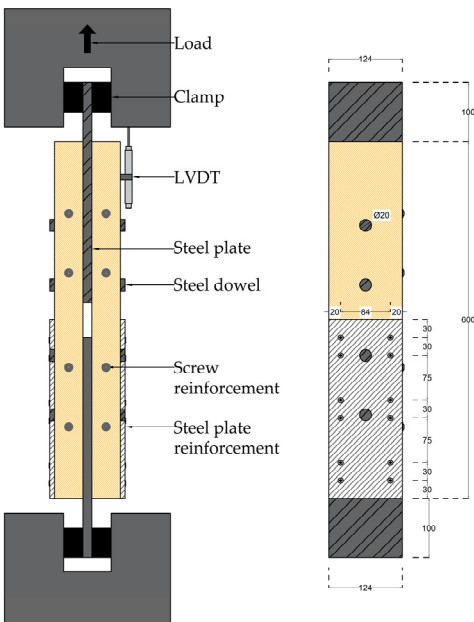

**Figure 4.** Specimen set-up (**left**) and dimensions for the specimen (**right**).

The upper part of the connection was clamped to the loading head and was pulled upwards during the test. The lower part had the same geometry as the upper part and the middle steel plate was clamped to the base of the loading machine. For the convenience of observation, this study intended to control the failure of connections to the upper part by reinforcing the lower part. For all connections tested in this study, both unreinforced and reinforced, additional steel reinforcement was screwed to the sides of the timber members on the lower part. Strain gauges were attached to the connections to measure their displacements.

A total of four groups, each with 10 repetitions, were conducted and their details are tabulated in Table 2. Figure 5 shows the screws used in this study.

**Table 2.** Summary of groups in the connection tensile test.

| Group | Description | Screw-to-Dowel Distance | No. of Specimens | Mean Density (kg/m³) (CoV) | Mean M.C.% (CoV) |
|---|---|---|---|---|---|
| UC * | Unreinforced | N/A | 10 | 459 (13%) | 10.7 (10%) |
| SNC | Reinforced by screw with 0% thread (N) | 1d | 10 | 459 (10%) | 11.0 (10%) |
| SFC * | Reinforced by screw with 100% thread (S) | 1d | 10 | 475 (13%) | 11.2 (12%) |
| SPC * | Reinforced by screw with 33% thread on the point end (BS) | 1d | 10 | 476 (12%) | 11.0 (12%) |

\* based on Zhang et al., 2016 [31].

## Screw type/Group assignment

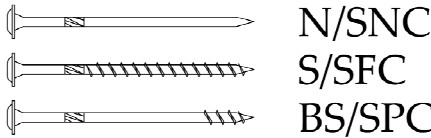

N/SNC
S/SFC
BS/SPC

**Figure 5.** Screw types and corresponding group assignment.

The connection specimens were loaded in tension with a constant displacement rate of 2 mm/min. The tests were stopped after a 20% load drop from the peak load had been observed.

## 3. Results

### 3.1. Embedment-Strength Test

The splitting failure occurred on all specimens in the embedment-strength tests. In most cases, a crack appearing and propagating below the dowel can be observed, as shown in Figure 6. Then, the specimens were cut open to investigate the deformation of the screws. In Figures 7 and 8, it can be found that screws with partial thread on the point end (groups S, BS, ES and TTS) tend to display a higher deformation together with a noticeable embedment of the screw head into the wood.

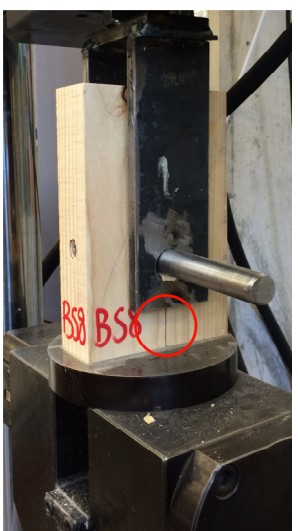

**Figure 6.** Camera captured the crack propagation on the surface of a reinforced specimen.

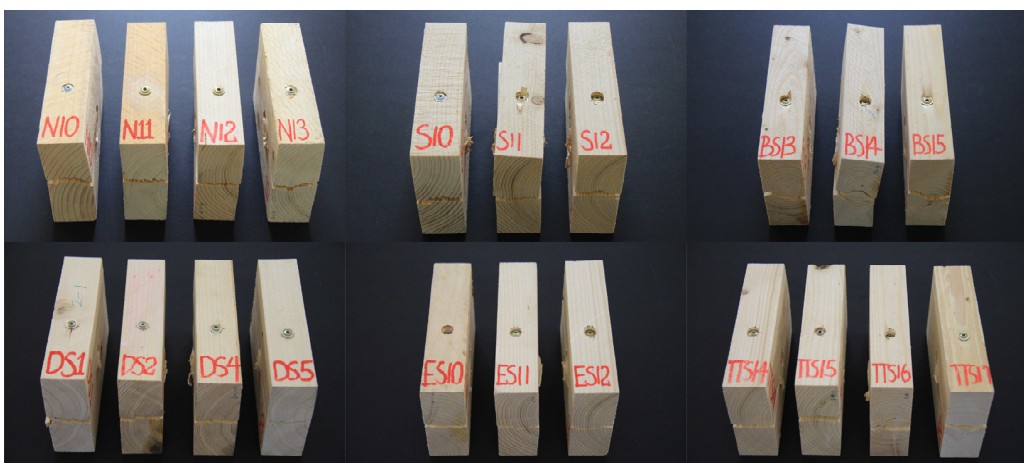

**Figure 7.** Embedment of screw head of a part of specimens from each group.

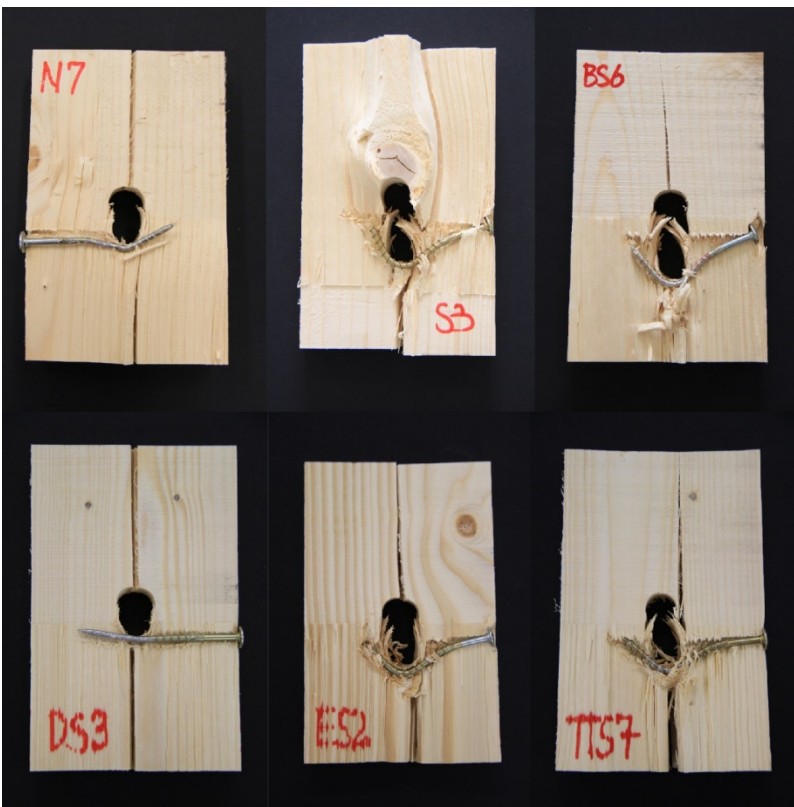

**Figure 8.** Specimens from each reinforced group after the embedment-strength test: high level of deformation of screw and embedment of screw head can be seen in groups S, BS, ES and TTS [31].

Based on test results and observation, reinforcement is effective when restraining forces are present on both ends of the screw. In other words, the thread–wood anchorage at the point end and the pull-through resistance from the screw head can restrain wood splitting. With a reduced splitting tendency, the bending strength of the screw can be utilised before the failure of the timber, resulting in a higher embedment strength. In contrast, a screw is unable to restrain the propagation of the crack if the thread on the point end is absent and a lower embedment strength can be observed. Figure 9 shows the typical load-displacement curves in this study.

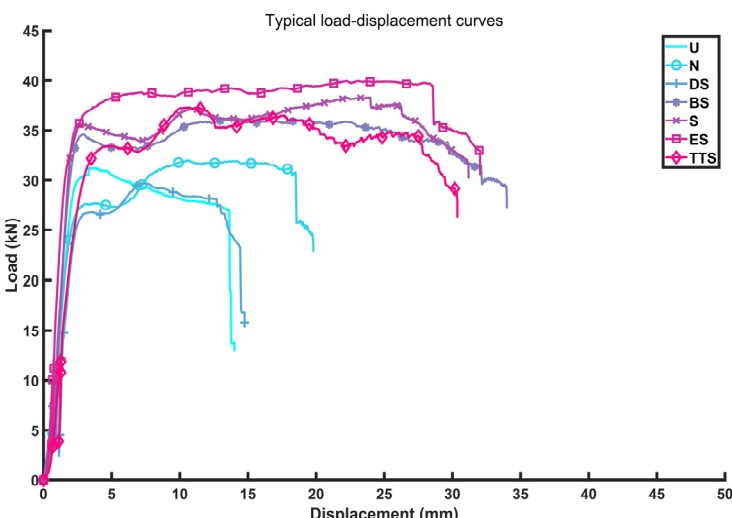

**Figure 9.** Load-displacement curves for embedment-strength tests.

### 3.2. Tensile-Connection Test

The load-displacement curves for each group are shown in Figure 10. Groups SFC and SPC show a higher load-carrying capacity and ductility than groups UC and SNC.

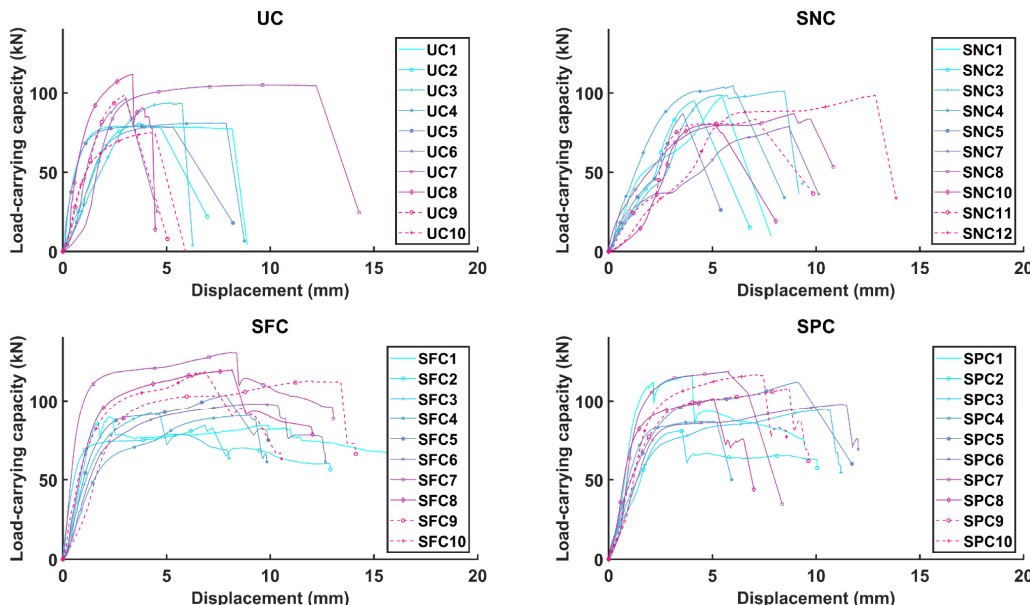

**Figure 10.** Load-displacement curves for each group in this study.

In the connection tests, splitting failure was observed to occur in the upper part of all specimens. After the splitting of the wood, the load acting on the connections dropped dramatically for the groups UC and SPC, while specimens in the reinforced groups SFC and SPC failed in a more ductile way with a gradual reduction in load. For most of the specimens in groups SFC and SPC, slight screw-head embedment was observed. The screws in these groups were bent by the dowels, as shown in Figure 11.

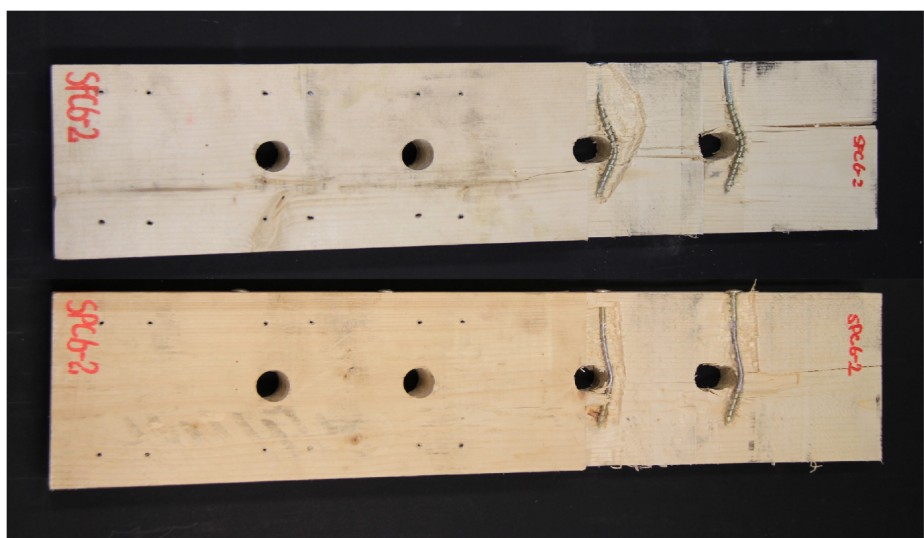

**Figure 11.** Timber members of connection specimens after failure showing deformation of screws. Top: connections reinforced by screws with complete thread (SFC). Bottom: connections reinforced by screws with 33% thread on the point end (SPC) [31].

Splitting failure occurred in most of the specimens in all groups, and shear plug failure was more prevalent for groups UC and SNC. In addition, the crack length in groups UC and SNC was generally longer than those in groups SFC and SPC, as demonstrated in Figure 12. This correlates well with the findings in previous work [26]: that self-tapping screws as reinforcement can control crack propagation.

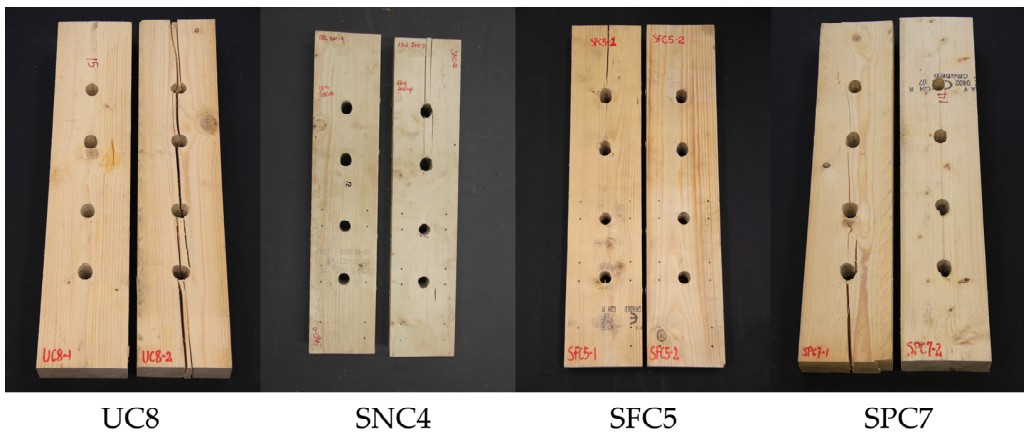

| UC8 | SNC4 | SFC5 | SPC7 |

**Figure 12.** Demonstration of specimen failures in each group.

## 4. Discussion

### 4.1. Embedment-Strength Test

The embedment strength of each specimen was obtained using the maximum load divided by the area of embedment by the dowel. Then, this study employed ANCOVA (analysis of covariance) to compare the results from different groups. The density of the samples was used as covariance. Table 3 presents the adjusted mean value and the enhancement ratio of each reinforced group compared to the unreinforced group.

**Table 3.** Results of embedment strength analysed by ANCOVA [31].

|  | U | N | S | BS | DS | ES | TTS |
|---|---|---|---|---|---|---|---|
| Mean embedment strength adjusted by ANCOVA (N/mm$^2$) | 31.05 | 32.15 | 35.6 | 35.48 | 32.22 | 37.53 | 35.31 |
| Enhancement ratio | 1.00 | 1.04 | 1.15 | 1.14 | 1.03 | 1.21 | 1.17 |

To identify the difference in effectiveness between each group, ANCOVA pairwise comparison was used to determine whether their difference was significant, with the results shown in Table 4.

**Table 4.** Significance of ANCOVA pairwise comparison for the embedment-strength test (groups U, N, S, BS, DS, ES and TTS) [31].

|  | U | N | S | BS | DS | ES | TTS |
|---|---|---|---|---|---|---|---|
| U |  |  |  |  |  |  |  |
| N | 0.481 |  |  |  |  |  |  |
| S | 0.004 * | 0.028 * |  |  |  |  |  |
| BS | 0.005 * | 0.034 * | 0.935 |  |  |  |  |
| DS | 0.452 | 0.963 | 0.031 * | 0.038 * |  |  |  |
| ES | 0.000 * | 0.001 * | 0.216 | 0.188 | 0.001 * |  |  |
| TTS | 0.001 * | 0.009 * | 0.652 | 0.594 | 0.010 * | 0.431 |  |

* Sig < 0.05 indicates a significant difference between the two groups.

Comparison between groups S and N showed that specimens reinforced by screws with 100% thread achieved significantly higher embedment strength than those reinforced by screws with 0% thread. This shows the effectiveness of a nail as reinforcement is limited and the importance of having thread on the screw.

In groups BS and DS, with the same 33% thread but at different locations (see Figure 3), the mean embedment strength of group BS with a partial thread on the point end is significantly higher than that of the group DS. The results indicate that the thread on the point end is more effective, as screws with 33% thread on the point end and with the flange head can form an anchorage action against the cracking of the wood. The screw was able to restrain crack propagation using the withdrawal capacity from the point end and the pull-through resistance from the screw head. Unlike the specimens in group DS, the specimens in group BS had not yet failed when the dowel was bearing on the screw; thus, the bending strength of the screw can be utilised, and a higher embedment strength was achieved.

In addition, group BS has significantly higher embedment strength than that of the un-reinforced group while group DS shows no significant improvement over the unreinforced group. This again confirms that to control crack propagation, restraints on both sides of the crack are required. Thus, group DS cannot utilise the bending strength of the screw to improve the embedment strength, as most of the specimens had failed before the dowels started to bear on the screw.

As for group TTS, using screws with 33% thread on both ends, Table 4 shows it has no significant difference to group BS and a significant difference to group DS, concerning embedment strength. This again proves the importance of thread location to the effectiveness of screw reinforcement.

With the screw placed closer to the dowel, group ES showed a higher mean embedment strength than group S, but the difference between them was not significant. With the dowels being placed in contact with the screws, it is presumed that crack initiation was delayed as a restraining force was available as soon as the dowels were loaded.

Ductility is an important factor in timber connection design. A ductile structure is preferable, as it can provide visual warnings of large deformations before a failure occurs.

In seismically active areas, a ductile timber connection can dissipate more energy during earthquakes, in order to reduce the damage to the structure.

Ductility from the embedment-strength test was calculated using two methods: the one shown in BS EN12512 [34] and the Forintek (FCC) summarised in Karacabeyli and Ceccotti [35]. BS EN12512 uses a secant and a tangent line to two sections of the load-deformation curve to determine the yield point, while the Forintek method utilises the point of 50% of the maximum load capacity to determine the yield point.

Table 5 summarises the average ductility of each tested group. As can be seen, the unreinforced specimens had the lowest ductility.

**Table 5.** Average ductility of each group calculated using two methods [31].

| Group | U | N | S | BS | DS | ES | TTS |
|---|---|---|---|---|---|---|---|
| Average ductility given by the EN 12512 method (CoV) | 8.1 (48%) | 15.4 (52%) | 21.4 (49%) | 22.4 (42%) | 9.4 (45%) | 19.3 (54%) | 11.5 (49%) |
| Average ductility given by the Forintek (FCC) method (CoV) | 12.2 (47%) | 24.6 (55%) | 32.0 (48%) | 34.8 (42%) | 14.8 (43%) | 26.6 (49%) | 15.6 (37%) |

Groups N and DS had lower average ductility than that of groups S, BS and ES. The screws in groups N and DS lack the ability to restrain crack propagation under loading and, hence, demonstrated brittle behaviour in test.

*4.2. Connection Test*

As the timber connections for each group were prepared from different batches of timber beams, the variation in density (see Table 2) may have resulted in different embedment-strength and load-carrying capacities. To reduce this effect, the load-carrying capacity of the connections were adjusted by ANCOVA, based on their corresponding timber densities. The mean capacity for each group, after adjustment of ANCOVA, is presented in Table 6.

**Table 6.** Results of load-carrying capacity adjusted by ANCOVA.

| | UC | SNC | SFC | SPC |
|---|---|---|---|---|
| Mean load-carrying capacity adjusted by ANCOVA (kN) | 92.2 | 92.5 | 103.6 | 102.2 |

The load-carrying capacity of groups SFC and SPC showed at least 11% increase compared to the unreinforced group, UC. The connections reinforced by screws without thread in group SNC only shows a slight increase, by 0.3%, when compared to the unreinforced connections.

Table 7 shows that the load-carrying capacity was significantly improved in the two reinforced groups (SFC and SPC) when compared to the unreinforced group (UC). The load-carrying capacity between groups SFC and SPC does not differ significantly. For group SNC, the difference to group UC is not significant and its mean capacity is significantly lower than that of groups SFC and SPC. Overall, there is good agreement with previous results from the embedment-strength tests. The connection tests also confirmed that the thread on the point end is key to maintaining the effectiveness of reinforcement.

**Table 7.** Results of pairwise comparison using ANCOVA.

| | UC | SNC | SFC | SPC |
|---|---|---|---|---|
| UC | | | | |
| SNC | 0.963 | | | |
| SFC | 0.018 * | 0.020 * | | |
| SPC | 0.037 * | 0.041 * | 0.756 | |

* Sig < 0.05 indicates a significant difference between the two groups.

### 4.3. Implementing the Embedment Strength of Reinforced Specimen in Connection Design

Currently, there are no available methods to predict the load-carrying capacity of screw-reinforced dowel-type connections. Embedment strength, $f_h$, can provide a path to predicting the theoretical capacity of reinforced connections. Based on BS EN 14358:2016 [36], the corresponding characteristic values of the embedment strength and the load-carrying capacity of connections from tests were calculated and are shown in Table 8. BS EN 14358:2016 assumes the test values to be logarithmically normally distributed, and the main value $\bar{y}$ and standard deviation $s_y$ is firstly determined using the equations below:

$$\bar{y} = \frac{1}{n} \sum_{i-1}^{n} lnm_i \tag{1}$$

$$s_y = max \left\{ \begin{array}{l} \sqrt{\frac{1}{n-1} \sum_{i=1}^{n} (lnm_i - \bar{y})^2} \\ 0.05 \end{array} \right. \tag{2}$$

$$m_k = exp(\bar{y} - k_s(n)s_y) \tag{3}$$

where

$\bar{y}$ is the main value;
$n$ is the number of test specimens;
$i$ is the i-th data point in ascending order;
$m$ is the material strength parameter;
$s_y$ is the standard deviation;
$k_s(n)$ is the factor used to calculate characteristic properties for testing;

When $n$ equals 10 and 15, the $k_s(n)$ is given as 2.1 and 1.99, respectively, in clause 3.2.2 in BS EN 14538:2016 [36]. Then, the characteristic values of the embedment strength can be found for each group.

**Table 8.** Characteristic embedment strength, characteristic load-carrying capacity and theoretical prediction [31].

|  | **U** | **N** | **S** | **BS** |
|---|---|---|---|---|
| Characteristic embedment strength from the embedment-strength test (N/mm$^2$) | 24.35 | 26.44 | 27.40 | 27.54 |
|  | UC | SNC | SFC | SPC |
| Characteristic load-carrying capacity of 10 connections (kN) | 69.46 | 73.68 | 77.87 | 77.49 |
| Theoretical prediction using the characteristic value of the embedment-strength test (kN) | 65.49 | 68.93 | 70.52 | 70.75 |

According to EC5, the characteristic load-carrying capacity for steel-to-timber connections is calculated as:

$$F_{v,Rk} = min \left\{ \begin{array}{ll} f_{h,1,k}t_1 d & (f) \\ f_{h,1,k}t_1 d \left[ \sqrt{2 + \frac{4M_{y,Rk}}{f_{h,1,k}dt_1^2}} - 1 \right] + \frac{F_{ax,Rk}}{4} & (g) \\ 2.3\sqrt{M_{y,Rk}f_{h,1,k}d} + \frac{F_{ax,Rk}}{4} & (h) \end{array} \right. \tag{4}$$

where:

$F_{v,Rk}$ is the characteristic load-carrying capacity per shear plane per fastener;
$f_{h,1,k}$ is the characteristic embedment strength in the timber member;
$t_1$ is the smaller of the thickness of side member or the penetration depth;

$d$ is the fastener diameter;

$M_{y,Rk}$ is the characteristic fastener yield moment;

$F_{ax,Rk}$ is the characteristic withdrawal capacity of the fastener.

The calculated characteristic embedment strength of the reinforced specimen has been substituted into Equation (4) to replace $f_{h,1,k}$ in the equation. The different calculation expressions in Equation (4) stand for various failure modes for a steel plate of any thickness as the central member of a double-shear connection. Failure mode (f) describes that only embedment failure occurs to the connection, failure mode (g) and (h) assume there are either one or two hinges formed in the dowel, respectively, together with embedment failure in the wood. The values from the theoretical predictions are lower than the characteristic values from the connection test results, see Table 8. This implies that the prediction is conservative and offers the possibility of developing a method to predict the load-carrying capacity of screw-reinforced connections.

Previous studies, such as [16], did not apply ANCOVA to include the influence of density, and, consequently, their results were scattered due to the variation in density. In this study, the influence of density was taken into consideration by ANCOVA, thus giving a more reliable comparison between group means.

The ductility of the connections has also been calculated using the method proposed by BS EN12512 [34] and the results are shown in Table 9.

**Table 9.** Average ductility of each group in connection test.

| Group | UC | SNC | SFC | SPC |
|---|---|---|---|---|
| Average ductility given by BS EN 12512 method (CoV) | 3.7 (54%) | 2.9 (49%) | 9.0 (52%) | 4.9 (29%) |
| Average ductility given by Forintek (FCC) method (CoV) | 5.4 (52%) | 2.9 (30%) | 11.6 (44%) | 7.1 (21%) |

Comparing the results from BS EN 12512 [34] and the Forintek (FCC) method [35] shows that the European standard is more conservative by showing a lower ductility.

## 5. Conclusions

The embedment-strength tests showed that thread configuration can influence the effectiveness of self-tapping screws as reinforcement. Embedment strength and ductility can be significantly improved when the screw is able to provide a restraining force on both sides of the crack by having the screw head on one end and a threaded part on the other end. Statistical methods cannot identify the significant influence of screw-to-dowel distance, though a higher average embedment strength was achieved by placing the screws in contact with the dowel.

The connection tests showed good agreement with the embedment-strength tests. This confirms that self-tapping screws can enhance the load-carrying capacity and ductility of connections as well as the potential to be used on modern glulam structures as a strengthening or a repair measurement. In addition, the results demonstrated that screws with partial thread on the point end achieved similar reinforcement performance to screws with complete thread. This implies that properly reducing the thread length has a positive effect on the installation of screws while maintaining the effectiveness of the reinforcement.

The results of this study are based on C24 European Whitewood (*Picea abies*) only, and the influence of screw design parameters such as the screw diameter and the type of screw head were not considered. Further studies with a larger sample size are recommended.

**Author Contributions:** Conceptualization, C.Z.; methodology, C.Z. and G.-J.S.; validation, C.Z. and G.-J.S. and X.-Y.L.; investigation, C.Z.; resources, H.-Y.H.; writing—original draft preparation, C.Z.; writing—review and editing, W.-S.C., X.-Y.L. and S.-D.X.; visualization, C.Z. and H.-Y.H.; supervision, W.-S.C., S.-D.X. and C.Z.; funding acquisition, C.Z. and S.-D.X. All authors have read and agreed to the published version of the manuscript.

**Funding:** This research was funded by the General Project of Science and Technology Plan of Beijing Municipal Education Commission [Grant No. KM202310005022].

**Institutional Review Board Statement:** Not applicable.

**Data Availability Statement:** All data, models, and code generated or used during the study appear in the submitted article.

**Conflicts of Interest:** The authors declare no conflict of interest.

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
