# Peer review of "Reinforcement of Timber Dowel-Type Connections Using Self-Tapping Screws and the Influence of Thread Configurations"

_forests, doi:10.3390/f14020409_

Round 1

Reviewer 1 Report

The paper “Reinforcement of timber dowel-type connections using self-tapping screws and the influence of thread configurations” is well-organised and the research is adequately presented. However, the laboratory tests in this paper were originally described by the Authors in the conference paper: Zhang, C.; Chang, W.-S.; Harris R. Investigation of thread configuration of self-tapping screws as reinforcement for dowel-type connection. In WCTE 2016, World Conference on Timber Engineering, August 22-25, 2016, Vienna, Austria. It is recommended that the Authors clearly indicate in the Introduction that the article was based on the conference paper. Please indicate which analyses are new in this paper and add citations to the figures and tables similar to or based on the ones from the conference paper. I recommend it for publication once minor corrections have been made

Reviewer 2 Report

The manuscript entitled Reinforcement of timber dowel-type connections using self-tapping screws and the influence of thread configurations is suitable for publication in the Forests journal once minor corrections have been made. The experiments presented in this paper were originally described in the conference paper (Zhang, C.; Chang, W.-S.; Harris R. Investigation of thread configuration of self-tapping screws as reinforcement for dowel-type connection. In WCTE 2016, World Conference on Timber Engineering, August 22-25, 2016, Vienna, Austria). For this reason, it is necessary to indicate that this article was based on the conference paper and to add citations to table titles and figure captions. Please revise your paper according to the comments below.

Reviewer 3 Report

The author revealed the reinforcement effect of screwing on the embedment strength and lateral capacity of dowelled joint with steel plate. The relationship between screw position and the reinforcement effect became clear from the test result. The theme is very local, but the result may contribute to someone’s research actions.

There are some unclear explanations in the manuscript as followings. The authors should modify them before publishing.

Page 4, Figure 4

There is no illustration about the dimensions for the specimen, which is different from the figure legend.

Page 7, Figure 9

From the graph, it is clear that the results of U, N, and DS are much lower than the others. However, the difference between them seen in Table 3 is not so large. There is a suspicion that the authors chose the specimens with lower embedment strength from U, N, and DS for an impression management. It is inappropriate in academic discussion.

Page 8, Lines 196-197

I couldn’t understand what the sentence means.

Page 9, Table 3

I guess that the embedment strength was derived using peak load obtained from the load-displacement relationship. The authors should be clearly explained it in the manuscript.

Page 9, Line 226

The report written by Karacabeyli and Ceccotti (reference [29]) used four methods for obtaining the yield displacement and ductility. However, only one ductility was shown in the manuscript. Which one the authors chose?

Round 2

Reviewer 3 Report

The authors sufficiently modified according to the reviwer’s comments.